# Beyond eves and cracks: An interdisciplinary study of socio-spatial variation in urban malaria transmission in Ethiopia

**Claudia Nieto-Sanchez**[1‡*], **Stefanie Dens**[2,3‡], **Kalkidan Solomon**[4], **Asgedom Haile**[5], **Yue Yuan**[6], **Thomas Hawer**[2], **Delenasaw Yewhalaw**[7,8], **Adamu Addissie**[4], **Koen Peeters Grietens**[1,2,3,4,5,6,7,8,9]

1 Department of Public Health, Unit of Socio-Ecological Health Research, Institute of Tropical Medicine, Antwerp, Belgium, 2 Witteveen+Bos Belgium N.V., Antwerp, Belgium, 3 Research Group for Urban Development, Faculty of Design Sciences, University of Antwerp, Antwerp, Belgium, 4 Department of Preventive Medicine, School of Public Health, Addis Ababa University, Addis Ababa, Ethiopia, 5 Ethiopian Institute of Architecture, Building Construction, and City Development (EiABC), Addis Ababa University, Addis Ababa, Ethiopia, 6 Faculty of Medicine, University of British Columbia, Vancouver, Canada, 7 Department of Medical Laboratory Sciences and Pathology, College of Health Sciences, Jimma University, Jimma, Ethiopia, 8 Tropical and Infectious Diseases Research Center, Jimma University, Jimma, Ethiopia, 9 School of Tropical Medicine and Global Health, Nagasaki University, Nagasaki, Japan

‡ CNS and SD shared first authorship on this work.
* cnieto@itg.be

**Data Availability Statement:** All relevant entomological and architectural data are included in the paper. The data supporting the qualitative

## Abstract

During the past century, the global trend of reduced malaria transmission has been concurrent with increasing urbanization. Although urbanization has traditionally been considered beneficial for vector control, the adaptation of malaria vectors to urban environments has created concerns among scientific communities and national vector control programs. Since urbanization rates in Ethiopia are among the highest in the world, the Ethiopian government developed an initiative focused on building multi-storied units organized in condominium housing. This study aimed to develop an interdisciplinary methodological approach that integrates architecture, landscape urbanism, medical anthropology, and entomology to characterize exposure to malaria vectors in this form of housing in three condominiums in Jimma Town. Mosquitoes were collected using light trap catches (LTCs) both indoor and outdoor during 2019's rainy season. Architectural drawings and ethnographic research were superposed to entomological data to detect critical interactions between uses of the space and settlement conditions potentially affecting malaria vector abundance and distribution. A total of 34 anopheline mosquitoes comprising three species (*Anopheles gambiae s.l*, *An. pharoensis and An. coustani complex*) were collected during the three months of mosquito collection. *Anopheles gambiae s.l*, the principal malaria vector in Ethiopia, was the predominant species of all the anophelines collected. Distribution of mosquito breeding sites across scales (household, settlement, urban landscape) is explained by environmental conditions, socio-cultural practices involving modification of existing spaces, and systemic misfits between built environment and territory. Variations in mosquito abundance and distribution in this study were mainly related to standard building practices that ignore the original logics of the territory, deficiency of water and waste disposal management systems, and

findings of this study/publication are retained at the Institute of Tropical Medicine, Antwerp and will not be made openly accessible due to confidentiality concerns as the dataset cannot be fully anonymised given the nature of the research. Data can, however, be made available after approval of a motivated and written request to the Institute of Tropical Medicine at ITMresearchdataaccess@itg. be/.

**Funding:** The study was funded by the Biotechnology and Biological Sciences Research Council (BBSRC) through a pump-prime funding (Grant number: BOVA004) from the BOVA Network (Building Out Vector-borne diseases in sub-Saharan Africa). The funders had no role in study design, data collection and analysis, decision to publish, or preparation of the manuscript.

**Competing interests:** The authors have declared that no competing interests exist.

adaptations of the space to fit heterogeneous lifestyles of residents. Our results indicate that contextualizing malaria control strategies in relation to vector ecology, social dynamics determining specific uses of the space, as well as building and territorial conditions could strengthen current elimination efforts. Although individual housing remains a critical unit of research for vector control interventions, this study demonstrates the importance of studying housing settlements at communal level to capture systemic interactions impacting transmission at the household level and in outdoor areas.

## Introduction

During the past century, the global trend of reducing malaria transmission has been concurrent with increasing urbanization [1]. Urbanization has traditionally been considered beneficial for vector control as urban environments typically limit mosquito breeding sites through redistribution of large-scale agriculture and increasing distance from open water bodies [2]. The adaptation of malaria vectors to urban environments, however, has created concerns among scientific communities and national vector control programs as malaria control has been mainly focused on rural populations and settings, particularly in African countries [3].

Urban malaria has been associated with clusters of people living near parks, water bodies, urban agriculture, or peri-urban fringes [4]. An increase in vector breeding sites in rapidly urbanizing areas has often been compounded by waste management problems linked to deficient public services and overcrowding [5], as well as accumulated waste leading to soil, air, and water pollution [6]. Artificial water containers, such as plastic bottle caps and discarded tires, are popular vector breeding sites for malaria and other vector-borne diseases, especially during rainy seasons [7, 8]. Urban agriculture expanding into peripheral belts and centres of many towns and cities in sub-Saharan Africa [9, 10], has provided ideal breeding sites in seed beds with shallow water, irrigation wells, cultivated wetlands or swamps, and even cattle and human footprints [11]. The process of urbanization and associated anthropogenic activities have forced malaria vectors to adapt to new breeding sites under potentially unexplored conditions, such as *An. gambiae* breeding sites in tree holes [12] and the recent identification of *Anopheles stephensi* in urban and peri-urban settings of Djibouti, Ethiopia, Sudan and Somalia [13].

Small-scale spatial heterogeneity further complicates the study of urban malaria transmission. Understanding the impact of built and natural environments at this level is particularly important, as improved housing conditions can play a critical role in breaking malaria's transmission cycle by reducing human-vector contact [2, 14–17]. Habitation conditions can affect transmission dynamics, as mosquitos are attracted by higher concentrations of carbon dioxide and other chemicals [18] and can easily circulate in spaces where human exposure is more focalized [19, 20]. Similarly, previous studies have determined that closing doors and eaves, as well as improving screening, ventilation and roofing systems can effectively prevent mosquitoes from entering domiciliary environments [21]. Implementing those measures through multi-sectoral strategies focused on environmental management, capacity building in the construction sector and adoption of prevention methods in at risk populations has been suggested as a way to increase sustainability of infrastructure-based measures. Additional difficulties are methodological in nature. In most cities of the world, it is not possible to establish clear cut differences between rural and urban areas because of the ongoing growth of urban spaces and the extension of rural lifestyles into urban contexts. Identifying priority areas to target prevention

efforts is also a complicated task, challenged by high mobility of urban populations that limit the capacity of national control programs' to localize transmission [22].

In Ethiopia, malaria is one of the most important health problems with 60% (nearly 52 million people) of the population at risk of infection. Three-quarters of the total land mass is regarded as malarious [23]. Malaria transmission is seasonal and unstable and mostly linked with the rainy seasons. The main transmission season follows the June-September rains and occurs between September and December, while the minor transmission season occurs between April and May following the February-March rains. As the peak in malaria transmission occurs during the harvesting period of the year, it has tremendous impact on the agricultural productivity and hence the economy of the country at large [24]. Ethiopia has made significant efforts to expand key malaria interventions throughout the country. Indoor residual spraying (IRS) using dichlorodiphenyltrichloroethane (DDT) was introduced in 1959; with the global malaria eradication campaign then different chemical insecticides have been used for malaria control [25]. Chloroquine was the first line treatment of all malaria species in Ethiopia before 1998. It was replaced by sulfadoxine-pyrimethamine (SP) after 1998 for the treatment of uncomplicated *P. falciparum* due to widespread decline in the efficacy of CQ [26].

Planning for scaling-up malaria prevention and control interventions started in 2003 with the support from the Global Fund to Fight AIDS, Tuberculosis and Malaria (GFATM) and in 2004, the Ethiopian Federal Ministry of Health introduced artemisinin-based combination therapy (ACT) as the first-line drug for treatment of *P. falciparum* malaria, along with rapid diagnostic tests (RDT), to improve diagnosis, and distribution of long-lasting insecticidal nets (LLINs) as main prevention method. Major scale-up began in 2005 with country wide distribution of RDTs, ACTs, LLINs and implementation of indoor residual spraying (IRS) [27]. Currently, Ethiopia has set the goal of malaria elimination by 2030. However, current vector control interventions, mainly LLINs and IRS, are becoming less effective due to insecticide resistance, outdoor transmission, and change in vector biting and resting behavior [28].

This proof-of-concept study aimed at developing an integrated methodology to further characterize social and environmental dynamics associated with housing and urbanisation potentially affecting risk of exposure to malaria vectors in Ethiopia. We contend that extending the unit of research from the individual home to the larger settlement and its urban landscape, can better explain the impact of building strategies (including home improvement) in malaria transmission. We took an interdisciplinary approach for this research based on the premise that social housing condominiums are massively built in Ethiopian cities following a standardized prototype that do not consider the context of the territory nor social dynamics impacting uses of the space, resulting in heterogenous forms exposure to vector-borne diseases for residents.

## Methods

### Study site

**This study was conducted in Jimma zone.** Despite reduced prevalence, malaria persists in this area, with *Plasmodium falciparum* and *Plasmodium vivax* as predominant parasite species [29]. Urban malaria in this region has been associated with weak health services, widespread economic disparity, and human mobility [30], as well as low levels of insecticide-treated net ownership and usage [31, 32]. Proximity to micro dams and to other stagnant water sources resulting from manipulation of the natural environment during urban expansion, has also been associated with malaria infection, particularly among children under five [30]. Clustering of *Plasmodium* infection has been observed at the household level in specific *kebeles* (administrative units) where control efforts are largely absent and where housing

structures present multiple open eves that facilitate mosquito circulation [29]. Following national guidelines, vector control strategies in the region are currently focused on universal distribution of LLIN, targeted IRS, and larval source management (LSM) as supplementary strategy. Plans towards elimination for the 2020–2025 period also consider strengthening distribution of antimalarial treatments, as well as collaboration with non-health sectors [33].

**The Integrated Housing Development Programme (IHDP).** The Millennium Development Goals called for action to address issues of environmental sustainability and poverty reduction. Rapid urbanization in Ethiopia—with a projected annual rate growth of 4.6% between 2015 and 2050 [34]—has been associated with persistent poverty, dependence on rain-fed agriculture, climate change, and the decline in readily available resources in the natural environment, as well as internal and external migration [35, 36]. In this light, Ethiopia aimed to shift toward poverty focused development strategies and infrastructure investment [37]. In an effort to address urban poverty and increase homeownership opportunities nationwide, the Ethiopian federal government initiated a large-scale public housing program, called the Integrated Housing Development Program (IHDP) in 2005 [38]. The initial goals of this program were to construct 400,000 housing units countrywide, expanding employment opportunities through construction jobs and enhancing the construction industry with the development of skilled labour and material suppliers [39]. The IHDP, and the Growth and Transformation Plans I & II that followed in 2009 and 2016 respectively, have focused on increasing the supply of owner-occupied housing for low and middle-income residents. These units are locally referred to as 'condominium' housing (i.e. multi-storied housing units for several households where communal areas are jointly owned and managed). Interested low-income residents enter a lottery and if selected, they can secure a unit with a 10 or 20% down payment based on income. Successful applicants pay only for the construction costs of their unit, while the government provides the land. Throughout the country, approximately 245,000 units have been constructed following this model [40]. Important pitfalls have been, however, identified in this form of condominium housing, including inhabitants' lack of appropriation over individual and common spaces, structural deficiency derived from poor design and construction problems, and deficient sanitation due to insufficient access to services such as sewage and electricity [41]. Previous studies have shown that in Jimma Town, in particular, IHDP has failed to reach most in need populations due to lack of consideration of the actual purchase power of low and middle income families in the region [42].

Three different IHDP sites located in Jimma Town were included in this study and are presented as Condominium D, Condominium A and Condominium Y (Fig 1). Each condominium had different types of apartments that could be classified in four typologies (M-1, M-2, T-4, T-9) that accommodate studios and 1, 2 or 3-bedroom apartments, as well as a bathroom with shower, bathrooms, and a kitchen. From design, all apartments should be connected to water, sewerage, and electricity systems (Fig 2).

## Study design

Entomology, medical anthropology and (landscape) architecture and urbanism were combined into an interdisciplinary study. Phase 1 contained preparatory work, including definition of work packages, setting up data collection tools, and drawing the base maps. Joint fieldwork was conducted during phase 2 from September to November 2019. Ethnographic data collection and mapping production were simultaneously carried out and complemented by entomological results. During phase 3 results were presented and integrated in face-to-face and online workshops. Architectural drawings and ethnographic data were superposed to entomological data to detect critical interactions between uses of the space and settlement

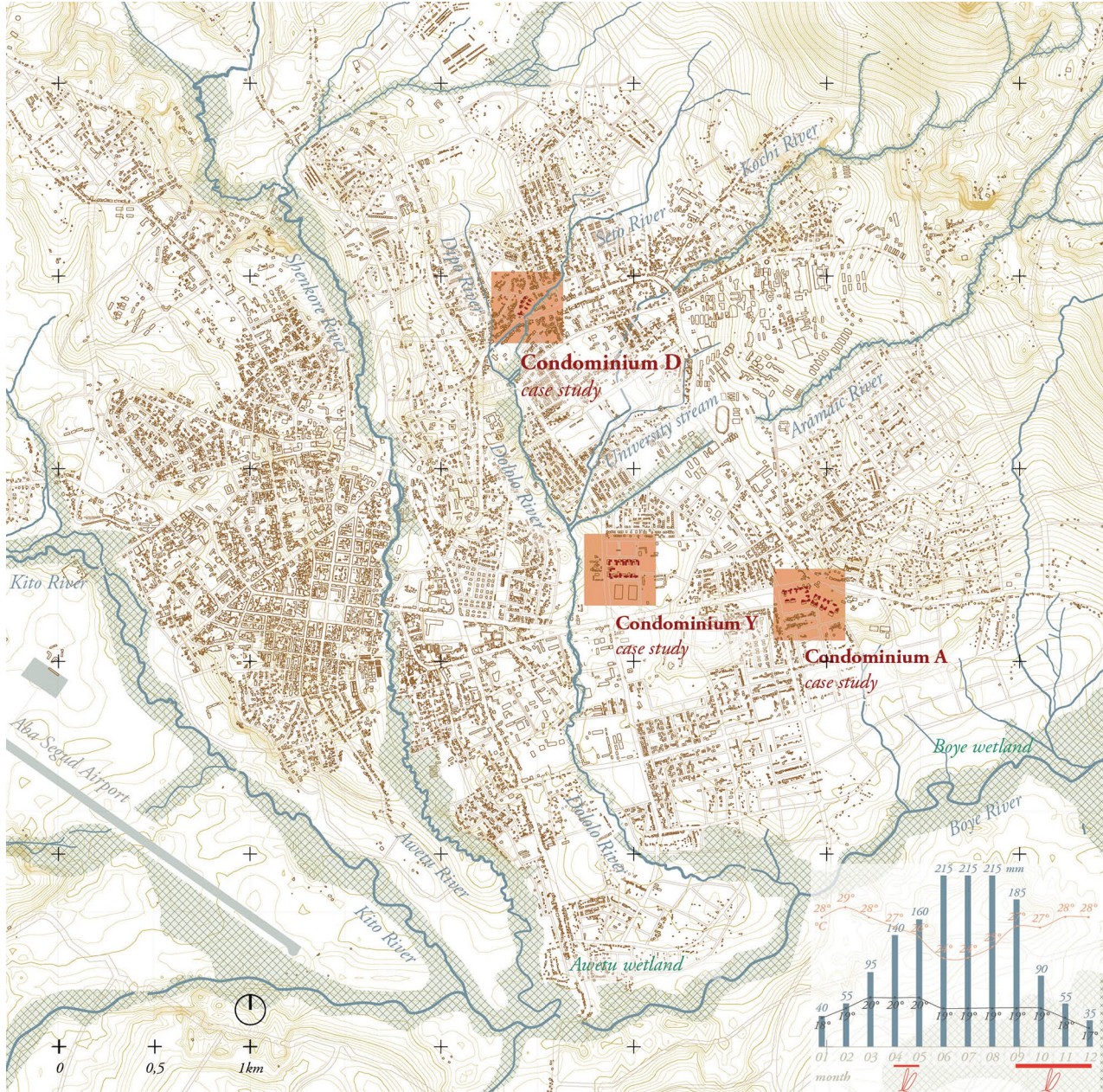

**Fig 1. Location of the researched condominiums along Jimma's floodplains.** Jimma is located in forested foothills and has an extended water network with wetlands in the south. Rivers are mapped in dry season situation. Bottom graphic shows monthly average rainfall (mm) and temperature (˚C) with indicated malaria transmission season. Map drawn by the authors, based on https://www.usgs.gov/ and openstreetmap.org.

context facilitating vector abundance and distribution in the selected condominiums. Phase 4 was dedicated to final data analysis and dissemination.

**Entomology.** *Data collection and mosquito sampling.* Three IHDP residential sites were included in this study. In each residential sites 3 blocks with ground, first and second floors were randomly selected. One house from each floor was selected for monthly mosquito collection using Centre for Disease Prevention and Control (CDC) traps for 3 months. In each month mosquitoes were collected both indoor and outdoor for two consecutive nights using

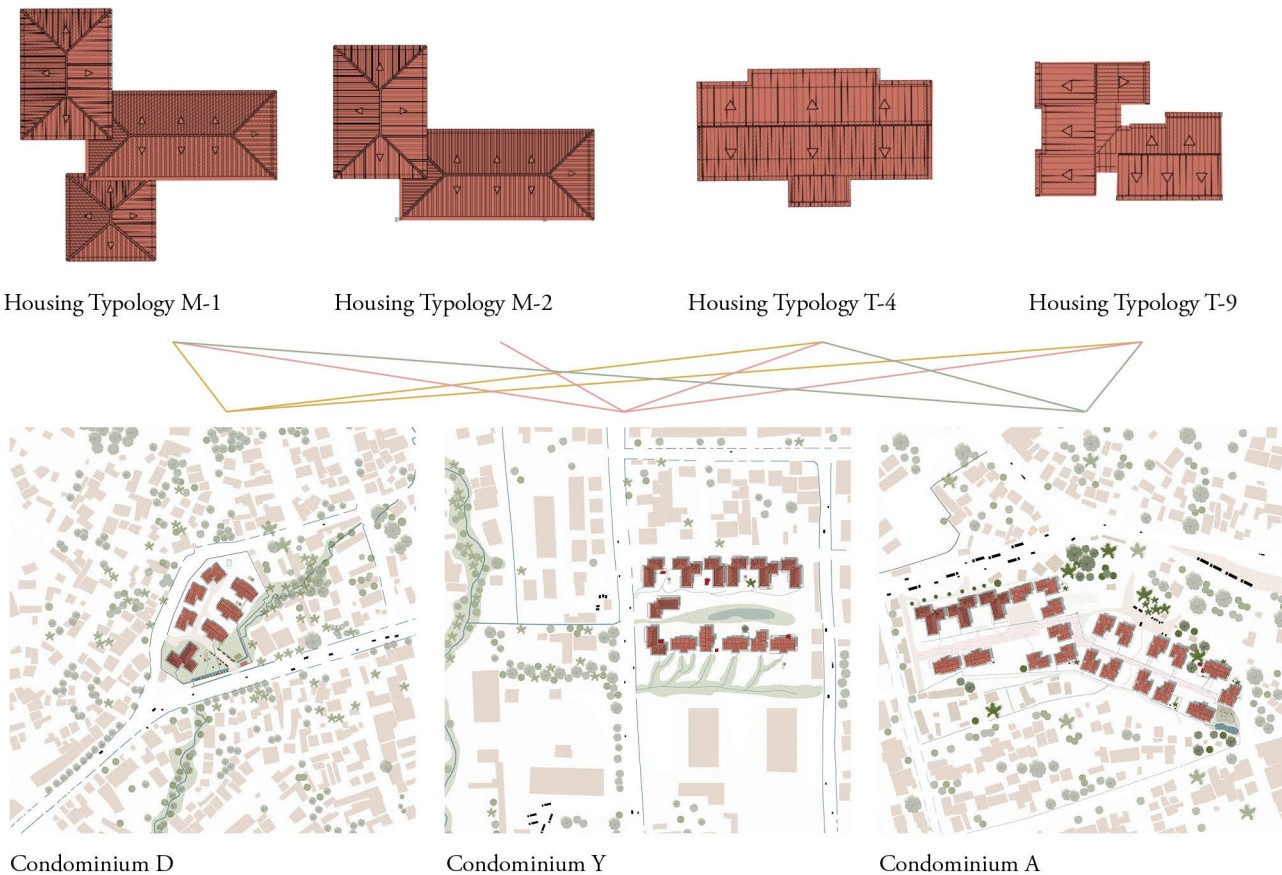

**Fig 2. Condominiums and standardised housing typologies in Jimma, Ethiopia.** Map drawn by the authors, based on fieldwork, https://www.usgs.gov/ and openstreetmap.org.

CDC light traps. Hence, in each month, the CDC LTs were set in three blocks on ground, 1st and 2nd floor for two nights. CDC light traps were set inside selected houses near an occupied bed at a height of 1.5 meter from 18:00 to 06:00 hr in the night to collect indoor host seeking mosquitoes. For the outdoor host-seeking mosquito sampling, CDC light trap was also set in the vicinity (within 8 meter) of sentinel houses from 18:00 to 06:00 hr. Accordingly, mosquitoes were collected for a total of 36 CDC-nights (18 indoor and 18 outdoor) from each site every month for three months which was 54 trap nights indoor and 54 trap nights outdoor per residential site. Thus, a total of 108 trap-nights per residential site and this would give us an overall total of 324 trap-nights for the 3 residential sites for the entire study period. N.B. [3 residential sites*3blocks*3floors*3months*2nights/house*2places of mosquito collection (indoor & outdoor)]. Mosquito collection started in September and continued through October and November, which is the major transmission peak malaria season in Ethiopia.

*Data analysis.* Data were entered into a spread sheet and analyzed using SPSS version 20.0 (SPSS, Inc., Chicago, IL). Mean indoor and outdoor mosquito density was determined using T-test. The mean difference in mosquito density collected from the different floors of the residential blocks and densities of mosquitoes collected in the different months was compared using ANOVA. P. value < 0.05 was considered significant during the analysis. The species composition, abundance, dynamics and distribution of female Anopheles mosquitoes were determined in all condominiums. Indoor and outdoor mosquito density (mean no of mosquito per trap per night) and mosquito density in ground, 1st floor and 3rd floor was analyzed.

Mosquito physiological state was also determined during the study period. Species belonging to *An*. *gambiaes* s.l were selected from each hourly indoor and outdoor collection and were dissected for ovary parity using Stereo Dissecting Microscope to separate nulliparous from parous females based on tracheal terminations.

**Medical anthropology.** *Data collection*. An ethnographic approach was conducted to collect relevant data on uses and appropriations of the space, processes and structures of social organization, and factors influencing the experience of residency in IHDP condominium housing (i.e., homes and communal spaces). In-depth interviews, following continuously evolving semi-structured topic guides were conducted with residents of the three sites included in this study. Informal conversations were also held with participants in everyday life settings. Finally, group discussions aimed at eliciting varied and contrasting points of views on the targeted topics of research were organized in all sites but were only possible in two of them. Data were organized to recreate *systems of settings* and of *activities* as taking place in observed households and condominiums at large. Activity systems [43] are constituted by the activity itself (e.g. cooking), how is it carried out, how is it combined with other activities, and the meanings associated by those involved in such performance. Interviews and group discussions were audio-recorded and transcribed; note taking was used in the case of informal conversations.

*Sampling*. We followed a flexible and iterative approach for theoretical sampling, i.e. each participant provided new information that lead to the next participant at different stages during the data collection process. The sample included residents of all three condominiums, as well as key informants linked to the IHDP and the construction sector in Jimma city.

*Data analysis*. Data collection and analysis was a concurrent, continuous, flexible and iterative process. This means that preliminary data were intermittently analyzed in the field after which further research was conducted confirming or refuting temporary conclusions until the data could be theoretically supported. Raw data were processed in their textual form and coded to generate analytical categories or themes for further analysis using NVIVO v12 software [44].

**Architecture and urbanism.** *Data collection*. Desktop research allowed for preparatory map making. Open-source GIS data and Google Earth Pro was used to draw and trace maps on the scale of Ethiopia, the Omo River Basin and the urban fabric of Jimma, and to spatially analyse the city and the landscape, including its hydrology and ecology. Complementary, fieldwork was carried out in the three selected condominiums and their surrounding areas, which lead to (i) tailoring the initial maps based on the (built) reality on site, (ii) annotating cultural practices (using QGis, Adobe Illustrator and Autocad).

*Sampling*. In addition to the sites, 24 housing units were visited and drawn 'as built' based on the available IHDP standardized design plans and sections (received in Autocad). From these 24 researched cases, a selection of 9 representative units (informed by entomological and ethnographic data) were further elaborated.

*Data analysis*. We applied Research-by-design [45], i.e. a methodology in which the graphical making and output of maps and drawings reflects the knowledge obtained through literature review and fieldwork (Fig 3). The research-by-design methodology was applied to analyse and identify (i) the localization of cross-scale potential vectors' breeding sites (i.e., from the scale of the household to the scale of the urban realm and the landscape); (ii) spaces where socio-cultural practices lead to high vector exposure; (iii) systemic misfits in the spatial environment leading to vector abundance.

**Ethical considerations.** All components of this study were reviewed and approved by the Institutional Review Board of the Institute of Tropical Medicine in Antwerp, Belgium (ref: 1241/18) and Addis Ababa University Addis Ababa University (ref: 081/19/2019). Written and

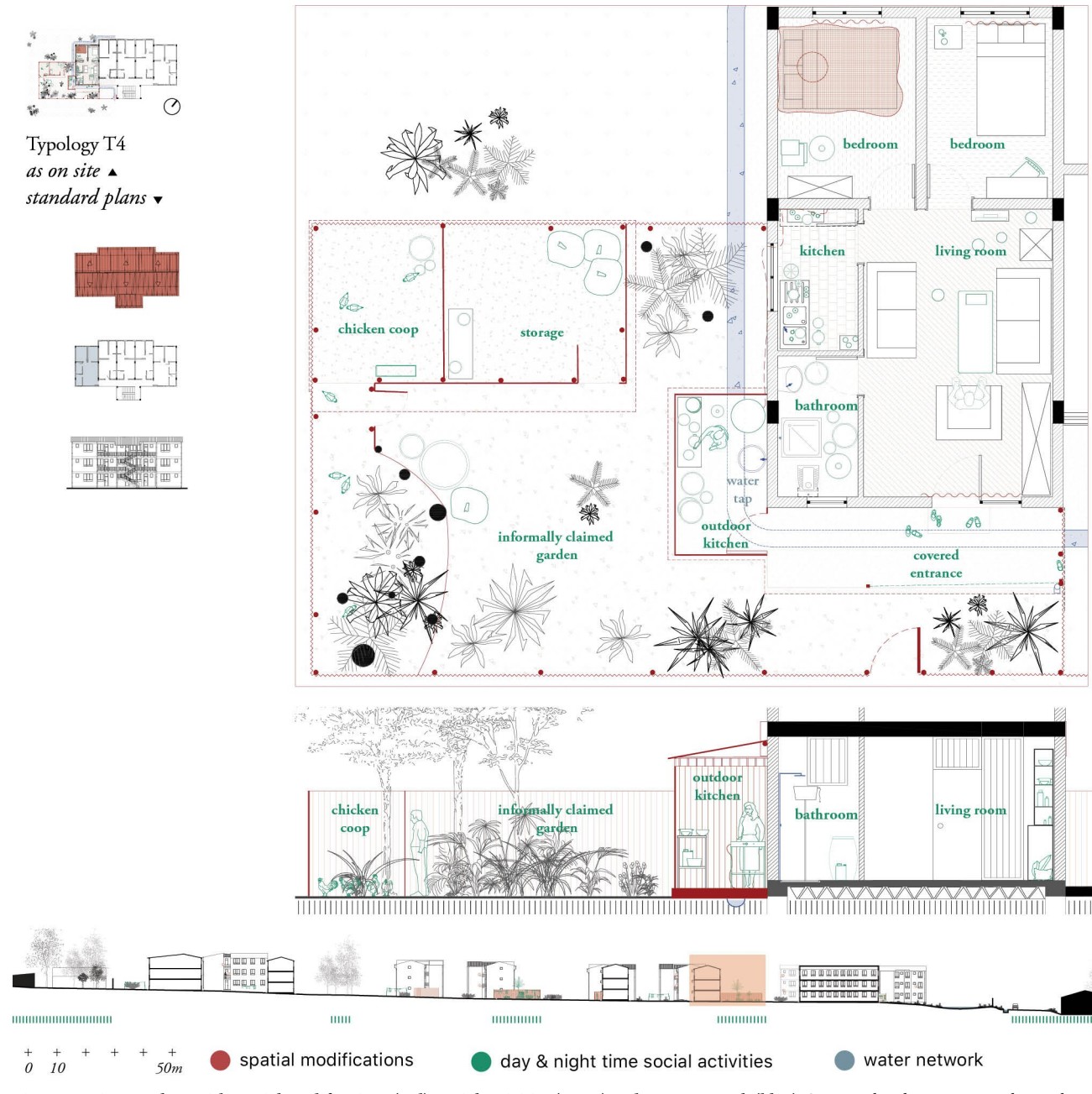

**Fig 3. Housing typology with spatial modifications (red), social activities (green) and water network (blue).** Structural and non-structural spatial modifications were mapped in red. They comprise informal housing extensions, added doors, removed walls, elevations and adjustments to the windows and other potential mosquito entry points. Spatial mosquito protective measures such as bed nets, curtains and other coverings of holes for screening purposes were also indicated in red,and complemented with socio-cultural activities and practices taking place at day (light green) and night times (dark green), such as cooking or laundering. These observations form a layer of appropriation of either the condominiums common spaces or the standardized housing plan. Finally, a blue layer was added to highlight the water network, including rivers, meanders, canals, and water taps.

oral informed consent was collected from all research participants. Study participants received an explanation about the purpose of data collection and their rights to drop out, stop the interview, or not answering specific questions. Collected data were anonymized and stored in password-protected devices to ensure confidentiality for participants. Pictures were altered to protect participants' privacy.

## Results

### Entomological data

A total of 34 anopheline mosquitoes comprising three species (*Anopheles gambiae s.l, An. pharoensis and An. coustani complex*) were collected during the three months of mosquito collection. *Anopheles gambiae s.l*, the principal malaria vector in Ethiopia, was the predominant species of all the anophelines collected. Apart from the anopheline species, 2,736 culicine mosquitoes were also collected. Majority of the anopheline mosquitoes (70.6%) and the culicines (66.1%) were collected outdoor.

Three of the total anopheline (9%) were fed (two *An. pharoensis* and one *An. gambiae s. l*), and the remaining 31 (91%) were unfed. More than half (53%) of the anopheline mosquitoes were collected from condominium Y site, followed by condominium D. Comparable proportion of culicine mosquitoes were collected from the three sites. Overall, the mean density of combined anopheline and culicine mosquitoes was 8.5 mosquitoes/trap-night, with significantly higher outdoor density t = 8.1, p < 0.001. Moreover, the mean density of the mosquitoes collected from the ground, first and second floor of the residential blocks was 13.4, 7.0 and 5.2 mosquitoes/trap-night, respectively. There was significant difference in mean density of mosquitoes collected from the different floors of the blocks [F(2,321) = 32.18, p < 0.001]. Pairwise comparison of the mosquito density from the different floor levels showed that highest mean mosquito density was obtained in the ground floor (p < 0.001). In the observed communal residential apartments risk of outdoor mosquito bite was higher than indoor.

The anopheline mosquitoes were collected from ground, first and second floors, with mosquito numbers sharply declined vertically from ground to the 2nd floor. More than half of the culicine mosquitoes were collected from the ground floor of the apartments. A similar declining trend of the number of culicine was recorded as the floor level increases from the ground to the 2nd floor. Distribution of the anopheline and culicine mosquitoes captured at different floors is presented in Fig 4.

The densities of mosquitoes collected in the different months is depicted in Fig 5. The densities of mosquitoes collected in September, October and November were 15.8, 4.4 and 5.4/trap/night. The difference in the density of mosquitoes in the three months was significant [F (2,321) = 78.6, p<0.001]. Pairwise comparison of the mean densities showed that highest mosquito density was obtained in September (p<0.001).

### Selected study sites

Condominium D hosts 7 buildings with 90 individual housing units that served approximately 250 residents. This site was administered by a local university and is the only gated site of the researched condominiums. Condominium Y is composed by 13 buildings, with capacity for 378 units and approximately 1.500 residents. Condominium A hosts 22 buildings, 318 flats and approximately 1.300 residents.

Study participants described condominium housing as a positive housing solution, particularly when compared to other government administered schemes in which toilets are shared and water supply and sanitation services are rarely available. However, spaces are generally characterized as sparce, particularly by large families, families using their home space for income generation activities, and those who have long term visitors or live with domestic help. Although initial designs considered inclusion of commercial units and communal spaces for social gatherings, none of studied condominium included any of these facilities. Original designs of kitchens did not include specific spaces for traditional ways of cooking on open fire

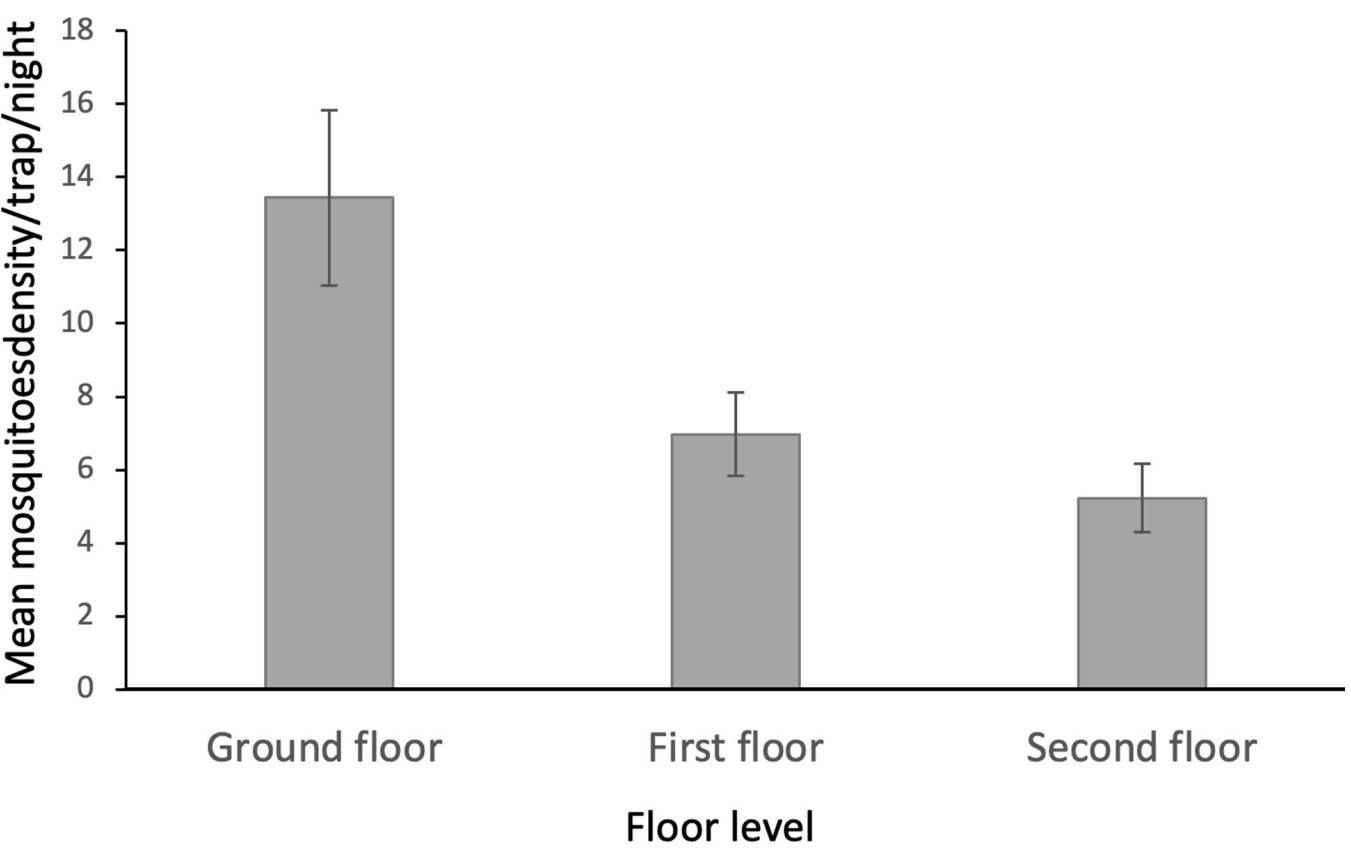

**Fig 4. Density of mosquitoes collected at different floor levels in all condominiums.**

—such as the ones employed for *enjara* (Ethiopian traditional pancake) production—or practices widely associated with major celebrations such as goat slaughtering during holydays.

Social dynamics, spatial arrangements and environmental factors affecting vector abundance and distribution create heterogenous forms of risk of exposure to vector borne diseases in the observed condominiums. We present these spatial variations grouped in three categories: landscape interactions, modification of artificial water networks, and standardized solutions for pluralistic lifestyles.

**(1) Landscape interactions.** IHDP condominiums were mainly constructed on available land owned by the government across the country. Of the observed sites, Condominium A is located near a large road, Condominium D is located in the in the floodplains of the Dipo River, and Condominium Y is located in the vicinity of the Dololo River In all cases, much more than adapting the architectural and urban design to the context, the context was manipulated to fit the standardised condominium designs (Figs 1 and 6). Alterations such as canalizing rivers, clearing vegetation, as well as elevating and flattening topography of the landscape destined to be construction site, were common in all the studied interventions.

Since Condominium D and Condominium Y are located in close vicinity of a river and constructed on lowlands, both were prone to flooding with water coming from the river and run-off water coming from surrounding higher urban areas during the rainy season. Condominium D was built directly in a floodplain, with the river itself canalized within the gates of the condominium that interrupting its meandering flow. Insufficient capacity of the canal leads to severe flooding issues during rainy and dry seasons. This situation is particularly

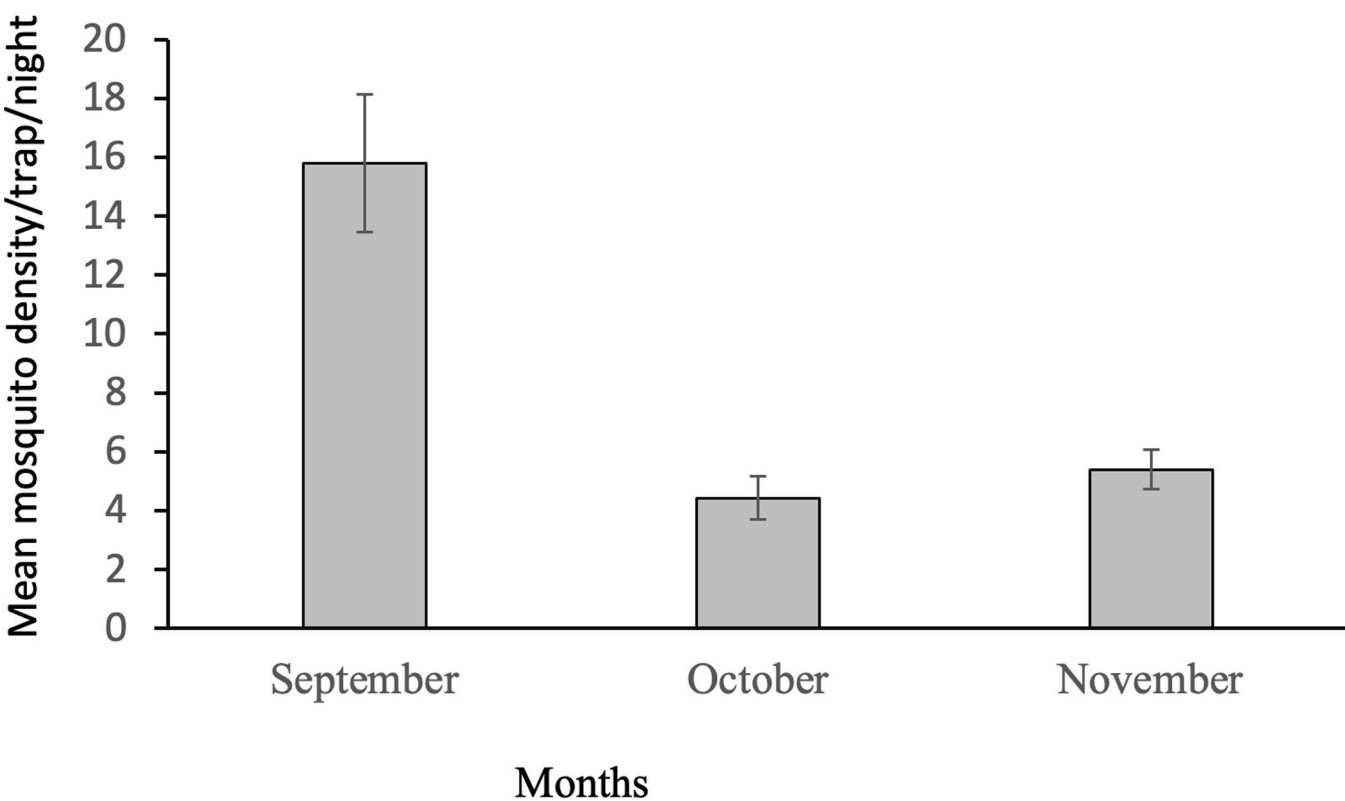

**Fig 5. Density of mosquitoes collected in different months in all condominiums.**

problematic in Condominium Y, as it was constructed on flattened land in the vicinity of the Dololo River (Fig 6). Its geographical location in the lower part of the city added to the topography of the site leads to permanent collection of stagnant water in the central grass field used as a playground. Abundance of mosquitoes was observed around these areas.

Jimma is known for its rich biodiversity and continuity of vegetation from mountain slopes through the city to the wetlands. However, the existing vegetation was cleared whilst flattening the land for the construction of the condominiums. The condominium settlement plans did not consider vegetation or landscape design, apart from paved surfaces for vehicles. Additional outdoor spaces were generally defined as 'grass land'. The settlement sections in Figs 1 and 6 show that the condominiums lack of vegetation forms a clear discontinuity in Jimma's forested landscape, which had been compensated by residents with individual gardens built on informally claimed land in the immediate vicinity of their households.

**(2) Artificial water networks.** In all three condominiums, the grey and black water system present systemic misfits, as the concept of the network was not tailored to each site specifically. Fig 7 illustrates the two artificial water networks observed on site: concrete gutters surrounding the housing blocks designed to catch rainwater falling from corrugated steel roofs make a rainwater network (left picture), while a grey and black water network was conceived as a subsoil piped network directing grey and black water from the individual units to the communal septic tank (picture on the right). In the case of Condominium D, part of the water was further directed into the Dipo River, turning the river into part of the sewage system. In the case of condominiums Y and A the grey and black water system ends in the septic tank, which lacks further connection to a larger sewage system in the near vicinity. Overflow happens on site.

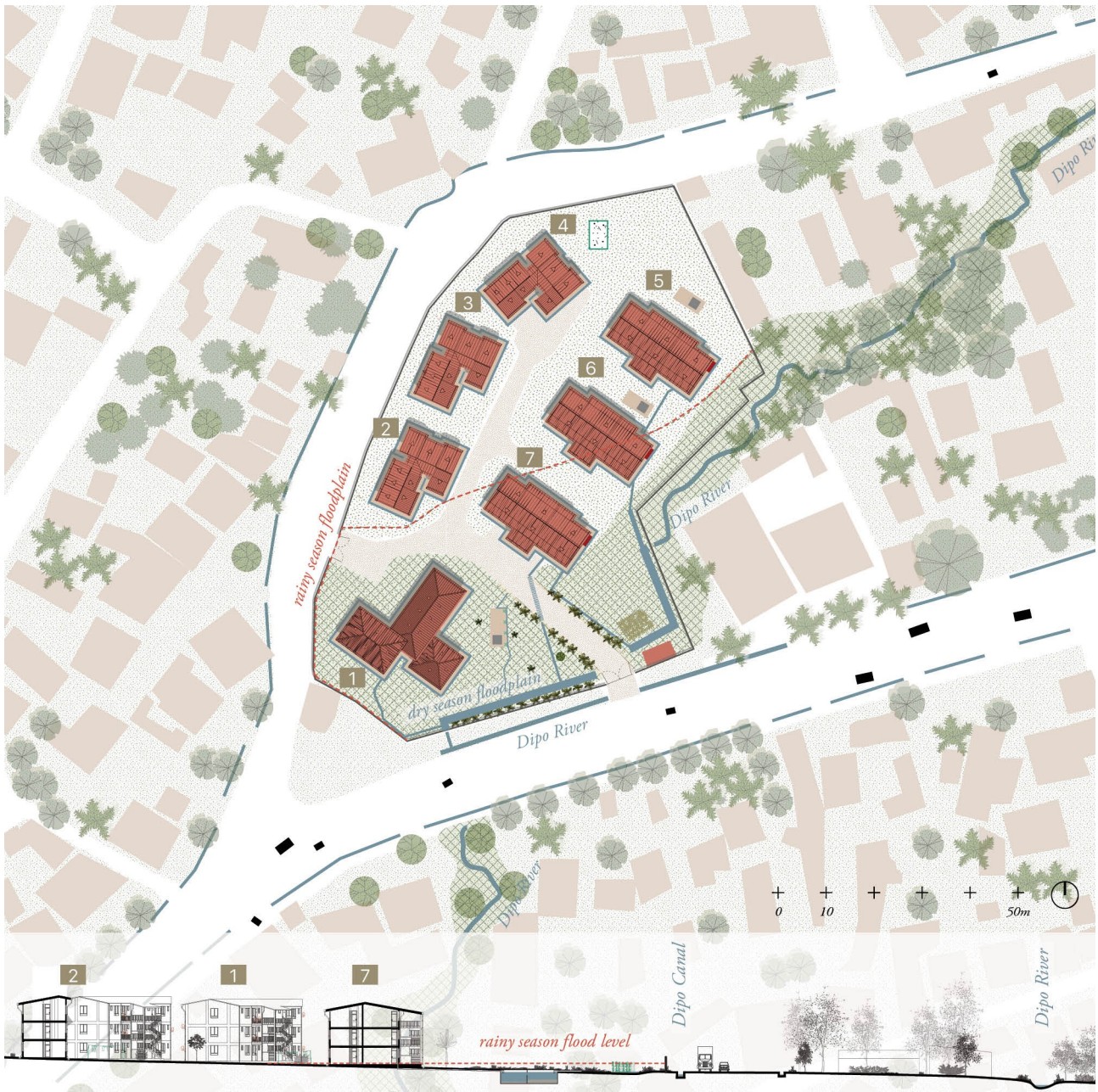

**Fig 6. Disruptive water networks in Condominium D.** Condominium D is located in the Dipo River floodplain, flooding the site moderately in dry season and extensively in rainy season (red dotted line). Within the walls of the condominium, the river is canalised and used as outlet for both the rain gutter network coming from the housing blocks and the septic tanks. The site suffers from permanently stagnating water. *Map drawn by the authors, based on https://www.usgs.gov/ and openstreetmap.org.*

Due to the condominiums' location on lower grounds, the grey and black water networks faced similar problems in the three sites. During the rainy season, groundwater tables were higher than the levels of black water in the septic tank, leading to overflow of the septic tank itself and backwashing of black water into the housing units. Similarly, during the rainy season or when the river waters present higher levels, the overflown septic tanks reach out to the river and pollute the entire site. It is unclear how the septic tank and the piped network were

maintained, as blockings of the network were found all the way from the housing units to the septic tank itself.

> *"The waste pipes and septic tank do not work properly. In order to keep sanitation in indoor areas, liquid waste from the kitchen and laundry are disposed to free spaces in the compound. But the septic tank is damaged and leaks because it has not been repaired so far. It cannot filter solid waste because the canals of our block are clogged and there is stagnant water. This makes the compound swampy, unpleasant, and perfect home for insects"* (Male)

Moreover, in all 3 condominiums these two artificial networks, designed to work in parallel, interfered with each other due to residents' solutions attempting to respond to, among others, the previously described water network issues. Three interactions potentially leading to increased vector presence were identified. First, gutters form a closed network of stagnant water prone to blocking and were not linked to a larger water basin from where the clean water could be reused. As a response, smaller ditches were dug, directing the water to the lowest point on site. In the case of condominium D, this lower point was the Dipo River, while for condominiums Y and A the ditches lead to swampy areas where water stagnates.

Secondly, due to clogging of the main internal piping network, residents had fixed alternative outlets through the wall that drains grey and black water from both kitchen and toilets directly into the rainwater gutter surrounding the housing blocks. As a result, rainwater was polluted by human waste, rendering the gutter into an open sewage system that often gets blocked by trash and produces heavy odours (Fig 7). The internal piping network shows permanent leakages, forcing the lower housing units to deal stagnant water under the leak and an altered internal piping network in which blockings were evaded by shortcutting neighbouring pipes. Backwashing of black water into the individual sanitation cell had been countered by residents by elevating the ground floor of toilets and showers.

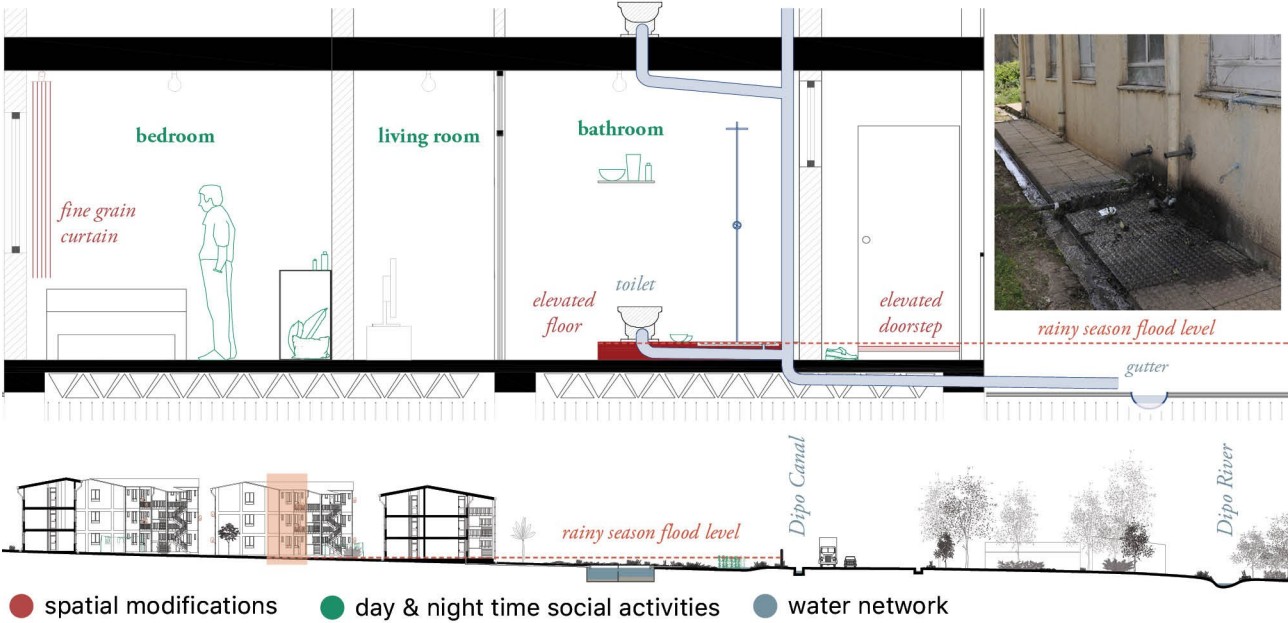

**Fig 7. Artificial water networks in Condominium D.** Detail of a typical altered plumbing pipes and ditch. Blocking of the original grey and black water networks leads to self-made solutions shortcutting on the original system. Black and grey water flow directly into the surrounding ditches.

Another expression of the limited capacity of water systems was the lack of water supply in the upper floors of the housing blocks when there was no electricity or when the demands in the lower floors increased. This creates the need to collect and keep water in different containers inside the homes for cooking and sanitary purposes.

According to research participants, the problems that lay at the base of the indoor drainage issues were the result of faulty water systems built with inadequate materials, as well as lack of technical knowledge to manage the demands of multi-storey housing. In addition, most of the septic tanks originally built in the condominiums had exceeded their capacity, which led to flooding and accumulation of stagnant water over green areas:

> "As I am the member of the committee, I know the problems in detail. In winter the houses are flooded. In general, there is a problem in its construction. I don't think it is constructed with the consultation of the expert." (Male)

A final point to explain the systemic nature of the problems associated with water management in the condominiums was waste accumulation. In Condominium D, solid waste was collected by residents at the home level and then transferred to a container on site, where persons hired for such purpose burn or transfer it again to other major dumping sites. Condominium Y and Condominium A (Fig 8), however, lack such an informal system: waste was thrown into

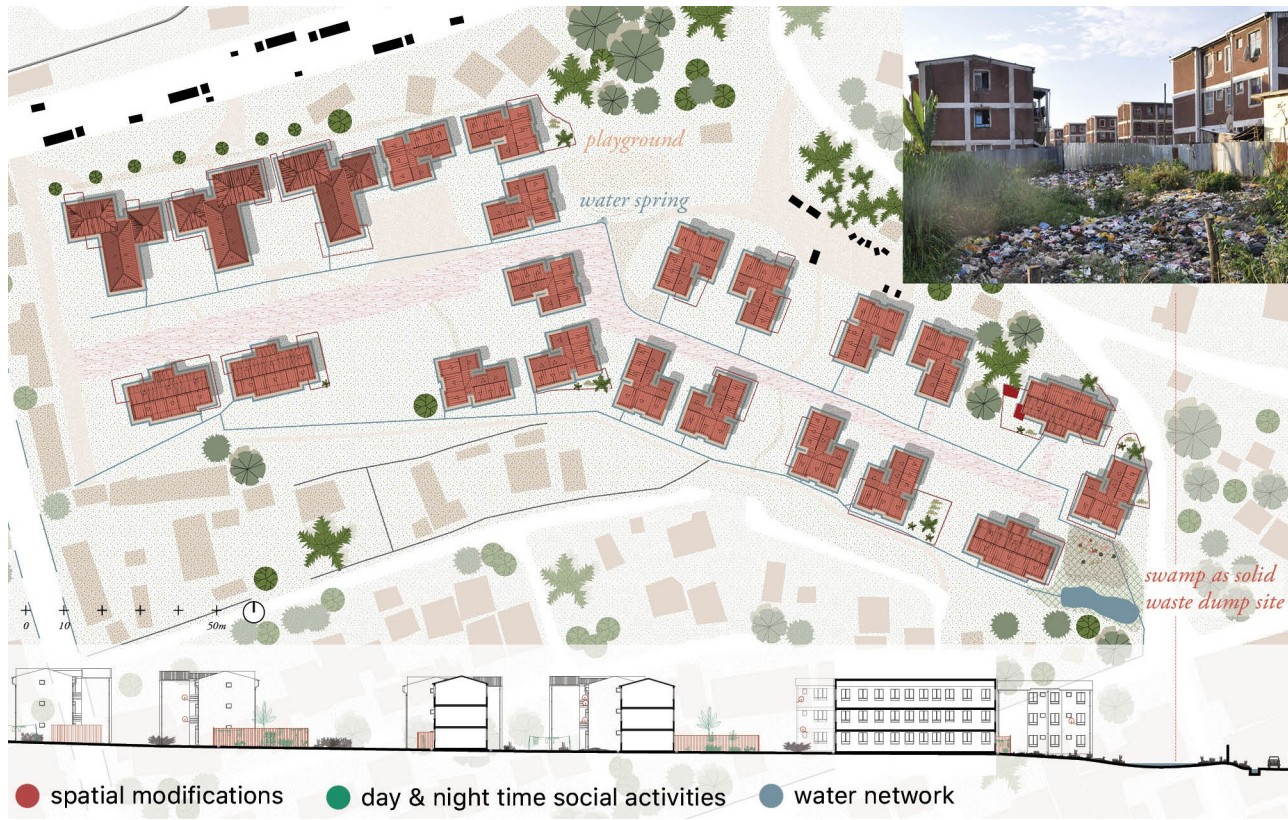

**Fig 8. Waste management in Condominium A.** The lowest point of condominium A is a swampy wetland, collecting rain, grey and black water. The area is used for waste collection, leading to a manyfold of potential mosquito breeding sites. The waste site collects all kinds of waste and is a habitat for chicken and cows. Next to this area, the excavation for the building site has cut through the ground water table, with a 'fresh' water spring as result. The spring is used for cleaning, washing, drinking and playing purposes. Graphic drawn by the authors, based on *https://www.usgs.gov/* and openstreetmap.org.

and next to the water networks, either directly or in the lowest point on site where solid waste swamps were growing as consequence.

*"Even if we are thankful that we are here, the house has its own problems. When we come to quality of life, there are many things that compromise quality of life, there are lots draining everywhere, there is a lot of mosquitos. When there is rain, it gets inside. . . the quality of the materials used inside the house is very poor, and because of that, there are always leaks and always needs maintenance. As you see, dry waste is dumped here in the compound so it's bad for the health of people living around there." (Female)*

**(3) Social appropriation of standardized space.**   Residents compensated the lack of formal spaces designated for cooking, laundry and social activities by informally claiming parts of the communal outdoor spaces. Informally claimed kitchens, kitchen-gardens and galleries were found in the condominiums. In ground floors, housing extensions are built by erecting physical barriers (with corrugated steel, for example), closing galleries or fencing gardens. In upper floors, families used metal sheets in verandas and corridors.

Most extensions have been built to serve cooking purposes. Not all units had a kitchen, and the ones included in the original designs, are narrow, with sufficient space to install electric stoves but not larger artefacts; therefore, they were mostly used for meals that do not require major elaboration and especially in the early hours of the day. To compensate the lack of kitchen space and to avoid air pollution indoors, halls and open areas were used to cook, particularly with charcoal. Other areas of the home were used for coffee making, as well as food storage (boxes, sacks and fridges). Cooking could also occur in common areas without a particular demarcation.

Another way of coping with the lack of space in housing units was adapting available space in the rooms for several purposes during daily routines. Participants described dynamic sleeping configurations, such as a studio that was used as a 'living' room during the day and 'bedroom' that accommodates a four-member family. Mattresses and blankets were extended over floors and furniture during the night and stored every day to maximize the available space for other purposes. The use of mosquito preventive measures was affected by this ongoing modification of the space. Families reported using bed nets and insecticide spraying—privately acquired or provided by the local government—as main insect protective methods applied at home. Our observations, however, indicated that the number of bed nets rarely matched the number of beds in each flat and in many cases, bed nets did not cover the entire surface they are intended for. That was the case for bunkbeds in which bed nets were suspended from the sides of the lower level and no protection was provided for the top level. In interviews, participants reported spatial concerns influencing their mosquito net selection. Although a rectangular bed net with 4 hanging points was more often recommended because it can cover larger surfaces, most families preferred a round one as it only requires one hanging point in the ceiling. Even though cylindrical bed nets fall short to cover the entire surface of the bed, they are perceived as more favourable.

*"No, we don't use bednet because the size of our family is big; there are some who sleep in just a mattress so we close all doors earlier in the day. Since my house is on the third floor there are no mosquitos. But we use Woiyra (plant used as natural insecticide)" (Female)*

Major spatial modifications of the standard housing units were carried out to facilitate income generation activities. That was the case of bars and restaurants built as extensions of housing units located at street level. In one of the examined cases (Fig 9), a veranda covered

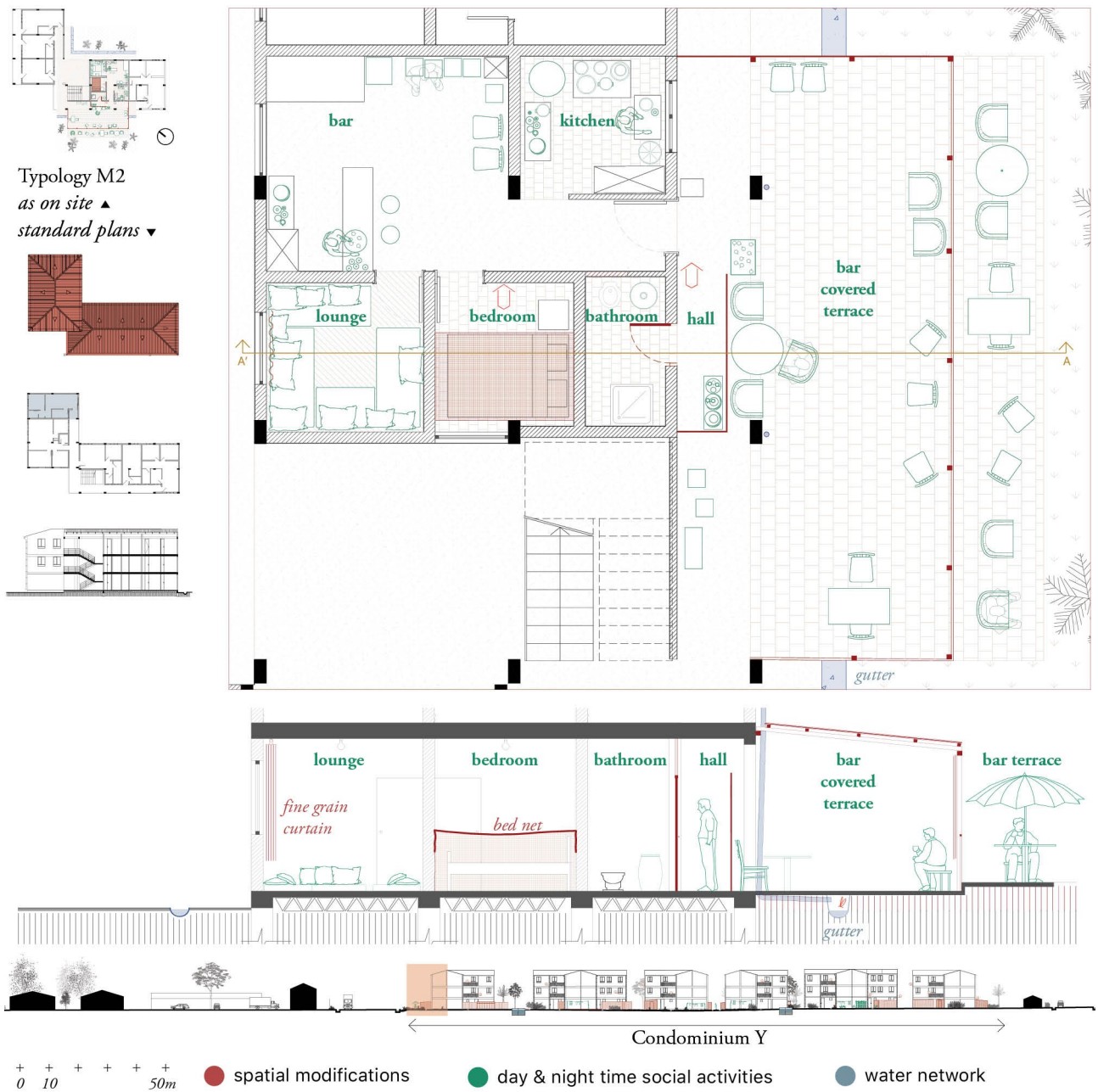

**Fig 9. Plan of the household with integrated bar.** The original apartment is extended with a covered terrace and floor tiles covering the gutter with stagnant water. Public and private activities are intertwined within this space. The adapted kitchen does not have access to running water. Windows are blocked and blinded, leaving the kitchen without any means of ventilation.

with corrugated steel roofing and light fabrics working as curtains extends the housing unit. The extension blocks the concrete gutter, leading to stagnant water around the apartment and its extension. Indoors, the housing unit had also been modified to meet the needs of the economic activities associated with the bar: bedrooms turned into kitchens and gathering spaces, as well as bathroom doors relocated towards the terrace. Although most residents had

extended their property in one way or another (appropriations can reach up to 1000 m$^2$), residents often expressed disagreement with this practice:

*"I don't feel comfortable with dwellers constructing individual fences because the concept of the condominium is to live together" (Male)*

Other reason to carry out larger modifications of the space was privacy concerns. Families implemented solutions that included closing the entrance to the bathroom through the living room and opening an alternative access through halls and common areas. In some cases, these modifications also implied construction of walls and relocation of toilets, mainly in ground floors, that could lead to disruptions of original water systems.

In the experience of research participants, the transition to condominium housing implied an adjustment of domestic routines, as well as of social interactions taking place around the home space. The specific location of the housing unit and the additional space that such location could potentially provide, the need to live in close proximity with multiple households, and the level of ownership over flats and common spaces, were often referred as factors positively or negatively framing their experience of inhabiting condominium housing. Frequent complains among neighbours were related to waste disposal in common areas, water filtering from upper to lower floors, demands for more cleanliness, as well as trespasses from private property.

*"It's very hard to say there is a social life here. When someone gives birth and when someone dies, people visit each other; but other than that, in this condominium we are almost the same people, we all have day work, we leave almost at the same time and come home at the same time, and everyone is busy with their child and so on, so there is no common gathering like drinking coffee together or visiting each other, that is not common." (Female)*

Residents reported mosquito presence in common areas where waste accumulates, as well as in their homes, especially during night-time when insects spread to the different rooms. Insects are particularly perceived as a nuance in spaces where residents conduct activities such as laundry, crops and spices' drying, celebrations, and kids' playing spaces. In general, participants associated the presence of mosquitoes with deficient sanitary conditions in the condominium and linked them with different health issues, including recent malaria episodes.

*"Vectors usually come in through the main door, the kitchen and also from the bedroom windows. They come from the direction of the toilet too since the stagnant water is there." (Female)*

*"Insects are very common in the night-time, since they want light in the day-time they can get it anywhere, but in the night-time they are attracted to the light bulb so they will get inside your home. They are mostly found around the liquid waste because they prefer that place and that liquid is available everywhere (. . .) I know malaria can happen here but I don't know the number so I can't say it is common or its not common." (Male)*

Besides individual uses of the space, original construction issues facilitated ongoing movement of vectors indoor and outdoor. It was common, for example, to find gaps between the window for door frames and the walls, easily allowing for vectors to enter the house. Paper and cardboard were used to stuff the gaps. Similarly, thick leftovers of cement plaster around the window frames prevented turning windows from opening, mainly in kitchens and bathrooms.

Residents responded to this obstruction by breaking the windows for ventilation purposes, and covering the broken glass with newspaper, cardboard or leftovers of bed net fabric. Where turning windows could be freely opened, improvised forms of screening designed by residents to prevent mosquitos from entering were observed. However, the lack of proper ventilation was mentioned as a general source of concern as units are often filled with bad odours that families identify as causes of respiratory and infectious diseases.

*"Because you cook lots of things in the house, ventilation is a must. But as you said, there are lots of windows that were left open and maybe broken [during construction], even mine are broken. They said they will repair them but that didn't happen for lots of reasons. So still opening the windows is important but there should be a wire mesh for insects not to get in. In the night-time it might get so hot and you might want to open the windows and sleep, so if there is a wire mesh then you will be safe. At the same time all pollutants would leave the house too. But if that is not possible, we can't afford to lose from both sides: being affected by the vector and at the same time being affected by poor ventilation." (Female).*

## Discussion

Residents of the observed condominiums were at risk of malaria infection due to the occurrence of both the principal (*An. arabiensis*) and secondary malaria vector (*An. pharoensis*) across sites, as confirmed by the presence of fed mosquitoes collected from indoor and outdoor areas. Although homes lay-out and construction practices remained important to explain mosquito abundance within the home space, extending the unit of research from individual homes to the settlement and its urban context, facilitated identification of systemic interactions that could sustain transmission even in outdoor settings. Systemic interactions creating variations in mosquito abundance and distribution in this study were mainly related to building practices that ignore the logics of territory ('genius loci') and its context, deficiency of water/ sewerage and waste disposal management systems, and adaptations of the space derived from the heterogeneous demands of residents.

Consistent with previous studies showing decreased mosquito densities in upper floors of multi-story buildings in Africa [46, 47], entomological results showed that inhabitants residing in ground floors were more likely to be at risk when compared to those residing in the first and second floors. Increased exposure on ground floors could be attributed to their constant proximity to ponds, gutters and ditches, sceptic tanks, flooding and other forms of stagnant water. In addition, since residents of ground floor apartments were more likely to use claimed areas for gardening and other forms or urban agriculture, it was also possible that vectors were brought closer to the home space, which could also enhance transmission.

In this study, the location of the observed condominiums was of particular importance to explain mosquito abundance. These plots were located at the edges of the city, where the characteristics of the territory reproduce—and in some cases, extend—the intersection between rural conditions and urban dynamics typical of growing cities. More than half (53%) of the anopheline mosquitoes collected were identified in Condominium Y, which is located on swampy lands, followed by Condominium D, located within the floodplain of an important river. Extended exposure could result from the collusion of these environmental factors and associated anthropogenic activity [48].

Our results suggest that housing settlements could be adapted to prevent the presence of breeding sites in malaria endemic areas. While modifications were multiple at the scale of the housing unit, little was adjusted at the settlement scale. Optimized gardening schemes and a

holistic vision on the water (and river) network could decrease vector abundance. For example, efficient black water (or sewage) and solid waste systems could be organized locally in a decentralized way on site, or, if the surrounding urban context already provides a centralized network, the site services could plug into that larger network. Observations on site in all 3 of the condominiums, however, showed that both the black water network, as well as the solid waste network, were designed and realized to plug into the larger centralized network of the neighborhood, but in all the observed cases this centralized network was non-existing. As a consequence of this misfit, residents built a manifold of ad hoc solutions, often leading to new misfits within the same network or relocation of the original problem.

Draining issues impact the condominiums at large and do so across scales, in a systemic way. Leaks and clogging issues affecting upper floors, as well as flooding and waste concentration in ground floors, create larger water disposal problems. The systemic impact of these interactions was even more apparent during the rainy season when floods, odours and mosquitoes affected all residents to some extent. Environmental management was another area that was likely to worsen over time. In malaria prone areas, deterioration of the land cover in addition to deficient waste management services and practices had shown a direct effect on mosquito abundance and composition [49].

As expected [50, 51], outdoor mosquito biting was higher than indoor. The activity systems organized around cooking, sleeping and social purposes hereby described, exemplify variations to traditional night-time activities that demand contextualized control measures. For instance, in the case presented in Fig 8, clientele can stay for relatively long periods of time in open areas during the most active hours of transmission. If these spaces were also located close to water bodies or among vector attractant vegetation, risk of exposure could also increase. In addition, these spatial modifications often alter spaces where water was used, where water should be prevented from circulating or where ventilation plays a crucial role, which again, could have an impact on mosquito abundance and exposure. Concurrently, radical transformations of the space for social and sleeping purposes decreases the possibilities of consistent and appropriate use of LLIN.

Similarly, alterations could create alternative forms of exposure beyond the ones assumed to emerge from improved housing, as design and construction problems that were not timely addressed become a permanent source of exposure for residents. This was exemplified in this study by sealed windows that prevented proper ventilation (particularly while cooking), as well as eves and cracks around windows and roofs originated in faulty construction practices. Breaking a glass to facilitate air circulation in fixed windows entails the risk of insect entry on regular bases. Moreover, materials used by families to reduce this exposure (paper, cardboard, or fabric) do not provide long lasting protection and, instead, could reduce awareness of potential risks. This study demonstrated that the protective nature of home improvement was closely linked to anthropogenic activity, as well as environmental modifications derived from the transformation of urban habitats in which housing solutions took place [49, 52]. Our results showed the protective capacity of building-related solutions (such as closing eves and cracks), could be limited if environmental interactions favoured vectors' abundance. Design-out-vectors becomes a necessary condition for successful building-out-vector strategies in urban contexts [21].

Finally, although IHDP condominiums were conceived as standardized social housing solutions to respond to increasing demands in large and medium size cities in Ethiopia, the relative high costs and financial mechanisms created to fund these projects had shaped condominium housing as a middle-class solution [42]. Issues of ownership should be considered when proposing specific control measures, as residents who were temporary tenants or renters of the

flats were less likely to be allowed or feel entitled to implement permanent solutions to infrastructure problems at their own cost.

This study was funded through a pump priming scheme. As a proof-of-concept study, it was mainly intended at providing methodological input for interdisciplinary studies focused on the intersection between living environments and infectious diseases. As such, mosquito population density and human exposure around the receptivity layers hereby identified demand further examination, particularly around the impact of vegetation and incipient forms of urban agriculture on vector abundance and distribution. Incorporating epidemiological data on actual malaria infection is necessary in future stages to determine the actual vulnerability of condominium residents and to better understand to emerging urban malaria transmission.

## Conclusion

Our results demonstrate the need to contextualize malaria control strategies in relation to vector ecology, social dynamics determining specific uses of the space, as well as building and territorial conditions. Importantly, although individual housing remains a critical unit of research for vector control, this study illustrates the importance of studying housing settlements at communal level to capture systemic interactions impacting specific transmission dynamics at the household level and in outdoor spaces. Our findings provided indications of how human activity can fundamentally alter ecosystems and landscapes and hence, transform or alter VBDs transmission dynamics even in apparently standard housing models. In urban contexts, architectural and urban design can substantially contribute to the realization of living environments conceived to interrupt VDBs transmission. Extending interdisciplinary studies and address systemic dynamics around living environments in urban areas remains of critical importance.

## Acknowledgments

We thank the staff from Jimma University Tropical and Infectious Diseases Research Center who were involved in mosquito and data collection. Also to research participants for their willingness to share their stories and spaces with us.

## Author Contributions

**Conceptualization:** Claudia Nieto-Sanchez, Stefanie Dens, Delenasaw Yewhalaw, Adamu Addissie, Koen Peeters Grietens.

**Data curation:** Claudia Nieto-Sanchez, Stefanie Dens, Kalkidan Solomon, Asgedom Haile.

**Formal analysis:** Claudia Nieto-Sanchez, Stefanie Dens, Kalkidan Solomon, Asgedom Haile, Delenasaw Yewhalaw, Adamu Addissie, Koen Peeters Grietens.

**Funding acquisition:** Claudia Nieto-Sanchez, Stefanie Dens, Delenasaw Yewhalaw, Adamu Addissie, Koen Peeters Grietens.

**Investigation:** Kalkidan Solomon, Asgedom Haile, Yue Yuan, Delenasaw Yewhalaw, Adamu Addissie.

**Methodology:** Claudia Nieto-Sanchez, Stefanie Dens, Delenasaw Yewhalaw, Adamu Addissie, Koen Peeters Grietens.

**Supervision:** Delenasaw Yewhalaw, Adamu Addissie, Koen Peeters Grietens.

**Visualization:** Stefanie Dens, Asgedom Haile, Thomas Hawer.

Writing – original draft: Claudia Nieto-Sanchez, Stefanie Dens.

Writing – review & editing: Claudia Nieto-Sanchez, Stefanie Dens, Kalkidan Solomon, Delenasaw Yewhalaw, Adamu Addissie, Koen Peeters Grietens.

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
