## [Decision Letter · Decision Letter 0]

5 Oct 2021

PGPH-D-21-00482

Beyond eves and cracks: An interdisciplinary study of socio-spatial variation in urban malaria transmission in Ethiopia

Dear Dr. Nieto, Claudia

Thank you for submitting your manuscript to PLOS Global Public Health. After careful consideration, we feel that it has merit but does not fully meet PLOS Global Public Health’s publication criteria as it currently stands. Therefore, we invite you to submit a revised version of the manuscript that addresses the points raised during the review process.

Academic Editors comments

While the reviewers agree on the importance of the topic, there is significant concern about the write-up, design, data collection, analysis, and interpretation of the data. There appears an urgent need for an additional level of subtlety in handling, analyzing and interpretation of data set. Reviewer #1 offers particularly detailed comments on how the authors may wish to approach revising the manuscript; overall, I advise very careful consideration of these and the other comments.

In my judgement, the manuscript requires major revision and re-review, with no guarantee that we can accept it for publication, as the changes are going to be substantial and challenging.

That said, it does seem that the authors are on the cusp of producing an important manuscript, so I would sincerely encourage a careful redraft and resubmission of the manuscript, and very much look forward to seeing it. I wish the authors all the best in their endeavours to meet the requirements laid out by the reviewers.

To facilitate further review and editorial feedback, please provide detailed notes as to how each reviewers' comment has been addressed in the revision.

We look forward to receiving your revised manuscript.

Kind regards,

Reginald Quansah, Ph.D.

Academic Editor

Journal Requirements:

1. We note that participants provided oral consent. Please state in the Methods:

- Why written consent could not be obtained

- Whether the Institutional Review Board (IRB) approved use of oral consent

- How oral consent was documented

For more information, please see our guidelines for human subjects research: https://journals.plos.org/plosone/s/submission-guidelines#loc-human-subjects-research

3. Since your data is not available for proprietary reasons, please explain via email why the data is not available. Please also include the contact information for the third party organization that should be contacted should other researchers want to request access to this data and please include the full citation of where the data can be found. We also request that you verify with us via email that any researcher will be able to obtain the data set in the same manner that the you have obtained it. If you feel you are unwilling or unable to adhere to this policy, please explain your reasons by return email and your exemption request will be escalated to the editor for approval. Your exemption request will be handled independently and will not hold up the peer review process, but will need to be resolved should your manuscript be accepted for publication. One of the Editorial team will be in touch if they require more information.

Additional Editor Comments (if provided):

PGPH-D-21-00482"Beyond eves and cracks: An interdisciplinary study of socio-spatial variation in urban malaria transmission in Ethiopia"

Reviewer 1

The concept is very interesting and coming of age. However, the manuscript falls short to deliver in explicit methodology especially in design, data collection and analysis of entomological data as well as other data. The abstract has information that is not found in the methods nor any other section of the manuscript. the abstract needs re-writing. The conclusion is not entirely supported by the data presented. data analysis was not explicitly described nor are the tools used for data analysis.

See more details below:

Line27? what progress?

27-28: what vectors? What out breaks? malaria?

31-33: this is not clear aim! what is the integrated methodology? (what is integrated)

31: This sentence is very confusing!

33: what is this unit? be clear

36: This should be the aim! No?

37: what study design was this?

47: abbreviation here is inappropriate!

Introduction: This introduction need to to be re-written. I understand the need to talk about the global and regiona situations on both VC and in Housing improvement, but what about Ethiopia? There is no information on the epidemiology of the disease, the control efforts put and also the NEED or GAP that this research findings will fill!

Where is the evidence of this? REF?

88: what is integrated?

90: We hypothesized that……………… this is unclear

93: I feel there is a need to expand on this as you have put in the abstract about Ethiopia adopting this type of housing.

112-116: since the aim is malaria transmission interruption, you should give justification for this work hence give a little background as to why these vector control tools are no sufficient for malaria control/elimination. I suggest this to be in the introduction

117-120: It is well to be information, but how is this linked to malaria transmission? i mean this can very well be in introduction if it is background information or is it the actual study site?

Also a map should be added for visual aid

132-133: so what were the combined study methods?

134-135: this is vague! what did you join? be specific

143-145: First, when exactly were these collections made? What months? what season?

The trap nights is confusing, you had a total number of 36 trap nights both indoor and out doors, 12 trap nights in 3 different floors.

Then you say a total of 108 traps for 12 trap night???

This is unclear because you have not explained your study site clearly!

how many sites? what was the unit of measure? houses? if so, what kind f houses are these? I do not see the need of having a section data collection and then a section of mosquito sampling!

combine the two and make give clear description of what was done, if need be, use figures for clarity

146: first time, write it in full

152-153: data analysis means explaining how you analyzed your data; parameters analyzed, how they were analyzed (if using software) or formulas, then what are those formulas? if its descriptive data, you say so. Bu this is lacking here, rather your are just giving explanations rather that analysis done!

154: 1st floor was not included? if so, why?

Also at the top you had mentioned only ground, 1st and 1nd floors! what changed?

155: What were those stages?

158-159: how was entomological innoculation rate calculated? what formula was used?

160: what do you mean?

also, before abbreviating anything, you need to write it in full

162-164: need to list the variables that were collected

179: how?

181: what were the variables?

200: Author to provide details such as the ethics approval reference number

205: on average per night, how many mosquitoes were collected? in the assumption tat your targeting malaria vectors, i think this would have been important to focus on Anopheles.

In the methodology section there was mention of physiological status, where are the results?

206: all those trap nights? when were these collections made? what season?

207-209: This a scientific paper suppose to have results that are scientifically sound, therefore, the need to have statistical significance in your results is paramount.

210-211: this is percentage... how about mean per collection night per position of trap (indoor Vs) outdoor)?

211-216: All these are just statements with proportions but with nos statistics to show the significance

217-218: how significant was the difference between floors and between species?

219-220: how significant was the difference between floors and between species?

450-451: what species?

453-456: Is the hypothesis proven? if so, then this needs to be said out right and by how much has it been proven (ie statistics)

462-464: no likely hood ration was calculated, hence how can you make such assumption? Need to calculate this ration for your statement to be true

464-465: in the FGD and interviews, was this mentioned? or in the study sites analysis, were these attributed mentioned or collected as pat of the study? if so, then it should be stated here and in results, if not, then look for supporting references

466-468: these are assumptions not proven by facts! either you have collected this data or you reference existing studies that have reported this.

469-470: Is this the first study to show the relationship between residential homes and proximity to mosquito breeding sites?

474-475: have this been reported elsewhere?

495-496: In the methodology, you did not specify where these traps were put both indoors and outdoors! you need to give these details

537-540: As pointed at the limitation of the study, is this truly demonstrated?

540-541: I doubt this has been effectively demonstrated by this study, i would say it calls for further research

Reviewer 2

Nieto-Sanchez and colleagues examined socio-spatial variation in urban malaria transmission in Ethiopia. Though descriptive in nature, the study provides an insight into the socio-spatial distribution of urban malaria transmission in Jimma which may help inform targeted interventions aimed at reducing the rate of malaria transmission.

See more details below:

Abstract:

The statements “Success in vector has been recently…” is not clear and needs to be revised for clarity and understanding.

Introduction:

a) The introduction provided by the authors is satisfactory. However, the authors could state clearly previous policies/interventions carried out in Ethiopia to mitigate malaria transmission in the country, especially in the urban cities, and how their current study will supplement those efforts in this regard. Part of this was rather noted in the methods section instead of in the introduction.

Methods:

b) What informed the authors choice of September to November for the data collection (fieldwork)? Could it be that it was the rainy or dry season or both? This should be explained.

c) What informed the authors selection of 3 sites, 36 trap-nights, and data collection over a period of 3 months? This warrants brief explanation.

d) Authors should provide citation for the NVIVO software in line 182 at page 9.

Results:

e) In line 215-216 at page 10, the statement “In the observed communal residential apartments risk of outdoor mosquito bite was higher than indoor” cannot be supported by the data as no such analysis was done to estimate and compare the risk of mosquito bite by indoor or outdoor status. The statement “…risk of outdoor mosquito bite was higher than indoor …” has a statistical meaning or the authors are just using it loosely? Thus, this statement should be removed or rephrased to reflect the kind of analysis done. Similar statements in other parts of the manuscript, including the discussion section should be rephrased.

Discussion and conclusion:

f) Again, statements like “entomological results showed that inhabitants residing in ground floors were more likely to be at risk of mosquito bites when compared to those residing in the first and second floor” in lines 462-464 at page 20 cannot be supported with the present data presented because no such formal analysis was done and presented. Could the authors show where they estimated such measures (e.g., odds ratios or relative risk or other similar measures) or they are using the terms like “… more likely to be at risk of mosquito bites …” loosely. Similar statements should be rephrased to reflect the kind of analysis done.

g) The authors may consider cutting extraneous text to improve the m

Reviewers' comments:

Reviewer's Responses to Questions

**Comments to the Author**

1. Does this manuscript meet PLOS Global Public Health’s publication criteria? Is the manuscript technically sound, and do the data support the conclusions? The manuscript must describe methodologically and ethically rigorous research with conclusions that are appropriately drawn based on the data presented.

Reviewer #1: Partly

Reviewer #2: Partly

2. Has the statistical analysis been performed appropriately and rigorously?

Reviewer #1: No

Reviewer #2: Yes

3. Have the authors made all data underlying the findings in their manuscript fully available (please refer to the Data Availability Statement at the start of the manuscript PDF file)?

Reviewer #1: No

Reviewer #2: Yes

4. Is the manuscript presented in an intelligible fashion and written in standard English?

Reviewer #1: No

Reviewer #2: Yes

5. Review Comments to the Author

Reviewer #1: The concept is very interesting and coming of age. However, the manuscript falls short to deliver in explicit methodology especially in design, data collection and analysis of entomological data as well as other data. The abstract has information that is not found in the methods nor any other section of the manuscript. the abstract needs re-writing. The conclusion is not entirely supported by the data presented. data analysis was not explicitly described nor are the tools used for data analysis.

Reviewer #2: Nieto-Sanchez and colleagues examined socio-spatial variation in urban malaria transmission in Ethiopia. Though descriptive in nature, the study provides an insight into the socio-spatial distribution of urban malaria transmission in Jimma which may help inform targeted interventions aimed at reducing the rate of malaria transmission.

See more details below:

Abstract:

The statements “Success in vector has been recently…” is not clear and needs to be revised for clarity and understanding.

Introduction:

a) The introduction provided by the authors is satisfactory. However, the authors could state clearly previous policies/interventions carried out in Ethiopia to mitigate malaria transmission in the country, especially in the urban cities, and how their current study will supplement those efforts in this regard. Part of this was rather noted in the methods section instead of in the introduction.

Methods:

b) What informed the authors choice of September to November for the data collection (fieldwork)? Could it be that it was the rainy or dry season or both? This should be explained.

c) What informed the authors selection of 3 sites, 36 trap-nights, and data collection over a period of 3 months? This warrants brief explanation.

d) Authors should provide citation for the NVIVO software in line 182 at page 9.

Results:

e) In line 215-216 at page 10, the statement “In the observed communal residential apartments risk of outdoor mosquito bite was higher than indoor” cannot be supported by the data as no such analysis was done to estimate and compare the risk of mosquito bite by indoor or outdoor status. The statement “…risk of outdoor mosquito bite was higher than indoor …” has a statistical meaning or the authors are just using it loosely? Thus, this statement should be removed or rephrased to reflect the kind of analysis done. Similar statements in other parts of the manuscript, including the discussion section should be rephrased.

Discussion and conclusion:

f) Again, statements like “entomological results showed that inhabitants residing in ground floors were more likely to be at risk of mosquito bites when compared to those residing in the first and second floor” in lines 462-464 at page 20 cannot be supported with the present data presented because no such formal analysis was done and presented. Could the authors show where they estimated such measures (e.g., odds ratios or relative risk or other similar measures) or they are using the terms like “… more likely to be at risk of mosquito bites …” loosely. Similar statements should be rephrased to reflect the kind of analysis done.

g) The authors may consider cutting extraneous text to improve the message.

6. PLOS authors have the option to publish the peer review history of their article (what does this mean?). If published, this will include your full peer review and any attached files.

**Do you want your identity to be public for this peer review?** For information about this choice, including consent withdrawal, please see our Privacy Policy.

Reviewer #1: No

Reviewer #2: No

---

## [Editor Report · Decision Letter 1]

23 Feb 2022

Beyond eves and cracks: An interdisciplinary study of socio-spatial variation in urban malaria transmission in Ethiopia

PGPH-D-21-00482R1

Dear Dr Nieto,

We are pleased to inform you that your manuscript 'Beyond eves and cracks: An interdisciplinary study of socio-spatial variation in urban malaria transmission in Ethiopia' has been provisionally accepted for publication in PLOS Global Public Health.

Best regards,

Reginald Quansah, Ph.D.

Academic Editor